# The Detection of Pine Wilt Disease: A Literature Review

**DOI:** 10.3390/ijms231810797

**Published:** 2022-09-16

**Authors:** Min Li, Huan Li, Xiaolei Ding, Lichao Wang, Xinyang Wang, Fengmao Chen

**Affiliations:** Co-Innovation Center for Sustainable Forestry in Southern China, College of Forestry, Nanjing Forestry University, Nanjing 210037, China

**Keywords:** *Bursaphelenchus xylophilus*, nematode identification, diagnosis of pine wilt disease

## Abstract

Pine wilt disease (PWD) is a global quarantine disease of forests that mainly affects Pinaceae species. The disease spreads rapidly. Once infected, pine trees have an extremely high mortality rate. This paper provides a summary of the common techniques used to detect PWD, including morphological-, molecular-, chemical- and physical-based methods. By comprehending the complex relationship among pinewood nematodes, vectors and host pine trees and employing the available approaches for nematode detection, we can improve the implementation of intervention and control measures to effectively reduce the damage caused by PWD. Although conventional techniques allow a reliable diagnosis of the symptomatic phase, the volatile compound detection and remote sensing technology facilitate a rapid diagnosis during asymptomatic stages. Moreover, the remote sensing technology is capable of monitoring PWD over large areas. Therefore, multiple perspective evaluations based on these technologies are crucial for the rapid and effective detection of PWD.

## 1. Introduction

Pine wilt disease (PWD) is a devastating forest disease caused by the pinewood nematode (PWN) *Bursaphelenchus xylophilus* (Steiner and Buhrer 1934; Nickle, 1970) [1]. PWD spreads quickly, and the mortality of the infected plants is extremely high; since the beginning of the 20th century, this disease has destroyed pine forest resources in infected areas. Due to human activities, *B. xylophilus*, an invasive species native to North America [2], has gradually spread to Japan [3], China [4], Korea [5] and other countries [6]; in these new areas, it has damaged forest resources and ecosystems [7]. Due to PWD’s rapid spread, wide range of hazards, and rapid onset, as well as the difficulties in preventing and controlling it, PWD is regarded as an imported quarantine disease in more than fifty countries worldwide [8,9].

Before *B. xylophilus* was established as the pathogen of PWD in the 1960s, most experts believed that the large number of pine tree deaths in PWD epidemic areas was caused by the feeding of pine beetles, as the dead pine trunks were observed to be almost completely covered with beetles, especially larvae of the family Coleoptera. However, a national project in Japan (1968–1971) implemented to control pine tree deaths, revealed that these beetles never attacked healthy pine trees, but only trees previously weakened by other biotic or abiotic factors; moreover, the affected pine trees wilted before insect attack [10]. In 1968, the pathologists Tokushige and Kiyohara observed the activity of suspected nematodes in the sawdust of dead pine trees under a microscope and identified them as belonging to the genus *Bursaphelenchus* [11]. In subsequent experiments, samples from several dead pine trees in Kyushu were collected, and nematodes were found in all samples. Then, a dead pine tree was selected, and the isolated nematodes were inoculated. The results indicated that these nematodes caused the death of pine trees, confirming their pathogenicity [12]. The nematode species was subsequently confirmed as *B. xylophilus*, and after large-scale culture and inoculation experiments, *B. xylophilus* was fully confirmed as the pathogenic factor of PWD [13].

After the PWN was discovered in imported scraps of wood from North America in 1986, *B. xylophilus* was added to the EPPO quarantine list [14]. *B. xylophilus* can infect most Coniferales plants, such as *Pinus*, *Abies*, *Cedrus*, *Larix*, *Picea*, *Pseudotsuga* and *Tsuga* species [4]. Additionally, while species in *Taxus*, *Juniperus*, *Sabina* and *Podocarpus* can be consumed by *Monochamus* beetles, there is no experimental evidence indicating that these tree species can be infected by PWD [4]. After infection, the following tree symptoms appear: first, the secretion of resin decreases significantly; second, the transpiration of the needles decreases and finally ceases; and finally, the needles of the entire tree turn yellow and then turn red but do not fall off that year. In the absence of human intervention, the trees will die once infected with PWD. The mortality rate is expected to peak in September, usually approximately 2 months after infection [15].

The first step in the prevention and control of PWD is to track the epidemic and diagnose infections quickly through quarantine and monitoring to prevent the disease from spreading to uninfected areas. Despite the quarantine measures, the pathogens may be introduced to new areas; thus, forests must be monitored so that the disease is detected in a timely manner [16]. In the first 20 years after the introduction of PWD to China, these measures were difficult to apply at the grassroots level, resulting in epidemic areas of thousands or even tens of thousands of acres because the disease was not detected and eliminated quickly. In the initial stage of PWD infection, pine trees do not exhibit symptoms, so it is difficult to accurately determine whether the trees in a forest are infected. Effective detection is a fundamental tool for pathogen monitoring and elimination; therefore, developing methods for fast, accurate and convenient disease diagnosis is urgent.

## 2. Direct Detection of PWD in Pine Trees

### 2.1. Diagnosis of Dead Pine Trees

Diseased pine trees often show the following characteristics: (1) the disease onset is from late April to early November; (2) the needles gradually lose their lustre and change from green to yellow, finally turning reddish brown without falling off the tree; (3) beetle oviposition sites are often found on the bark; and (4) when part of the bark is scraped off (e.g., with a knife), no resin flows out. According to the above characteristics, it can be preliminarily judged that pine trees are infected with PWD [17]. In April and May of the second year after infection, the symptoms of *B. xylophilus* in terms of the color of the needles are similar to those of diseased pines in autumn and winter. At this stage, when part of the bark is scraped off, resin outflow is observed. Additionally, while discoloration of the pine needles due to drought starts with the new upper needles and spreads to the older needles, the discoloration of PWD-infected trees begins in the old needles on the lower part of the tree and then spreads to newer needles on the upper part of the tree.

The above method of detection is suitable for a general survey of forest diseases. However, it requires trained, experienced personnel familiar with local forest facies and the symptoms of plant diseases and insect pests for good assessments; this method is largely subjective and is prone to misdiagnosis or missed diagnosis. Therefore, this method is usually used for a preliminary assessment of PWD and applied along with a pathogen identification method for improved accuracy and universal application.

### 2.2. Diagnosis of Standing Trees by the Oleoresin Exudation Method

Resin secretion is observed by punching a round hole with a diameter of 10~15 mm through the xylem of the trunk at breast height. Based on the ability of pine trees to secrete resin, the resin flow is classified into five grades: primary flow, characterized by resin outflow from the hole and a large amount of exudate; secondary flow, characterized by resin outflow from the hole and some exudate; third-stage flow, characterized by resin deposited on the lower edge of the hole; fourth-stage flow, in which granular resin is exuded from the hole wall; and fifth-stage flow, in which there is no resin outflow from the hole wall. According to the results of previous studies, the third-stage resin flow is a marker for PWD infection [18].

The oleoresin exudation method was first applied to facilitate the early diagnosis of PWD in forests. The method is characterized by its simplicity, easy application, and easy diagnosis protocol. However, many other factors affect resin flow in pine trees, hampering the accuracy of diagnosis using this method. Additionally, it is difficult to treat pine trees once the resin flow stops (i.e., fifth-stage flow). Therefore, this method has been used only as an auxiliary diagnostic tool.

## 3. Detection Based on Pinewood Nematode Morphology

The identification of PWNs based on morphological characteristics under a microscope is a traditional PWD detection method. The nematode is identified based on body length, tail length, tail tip shape, width of the middle esophageal ball, position of the vulva and the presence or absence of a vulval cover [19,20,21]. The main morphological characteristics include high lips and a narrowed head; females have a rounded tail, and some have a short mucro and flat vulva, while males have spicules curved with a cucullus [2,22]. Morphological detection is simple and easy and does not require complex instruments or equipment; thus, it is the most widely used method in production and laboratory research [23]. However, *B. xylophilus* is morphologically similar to *B. mucronatus*, which is non-pathogenic [24,25,26]. The identification can also be subjective, especially in the case of overlapping morphological features. Moreover, it is difficult to determine the species of juveniles and male adults by morphological methods; therefore, morphological detection requires personnel skilled in nematology with professional experience. Morphological detection takes a relatively long time, and when there are no typical female adults in isolated samples, the samples must be observed separately.

## 4. Detection of Pinewood Nematode DNA

Because of the increasing circulation speed of the wood market, the morphological identification method is far from meeting the demand. In recent years, with the continuous maturation and development of molecular biology technology, molecular detection methods for the PWN have been established and improved, and their application in practical work has become increasingly extensive [27]. This section focuses on a variety of molecular detection technologies developed based on DNA and analyses their respective technical characteristics and application in the field of PWN detection, aiming at providing a reference for the development and application of new rapid molecular detection technologies for the PWN, which has very important theoretical significance.

### 4.1. Restriction Fragment Length Polymorphisms (RFLPs)

Prior studies have used the RFLP technology to distinguish *B. xylophilus* and other PWNs via the 5.8S gene [28], the combination of the *Dra* I and PvuII restriction endonucleases [29], the *Hin*f I restriction endonuclease [30], and the *Dra* I and *Sal* I restriction endonucleases [31]. The RFLP technology is an important tool for analyzing genetic variation within and between nematode species. It has potential use in the identification of nematode populations and the study of genetic relationships among populations. However, this method is complicated and has a long operational time. In addition, it requires the extracted nematode DNA to have a certain level of purity.

### 4.2. Polymerase Chain Reaction (PCR)

Since its inception, PCR has become one of the most commonly used and important molecular biology techniques [32,33]. PCR revolutionized basic research on the molecular diagnosis and pathogenicity of plant parasitic nematodes. Molecular diagnosis based on sequence differences has become a general approach to plant pathology, with several applications: phylogenetic analysis, provenance identification and pathogenic gene exploration. Different PCR primers are used for pathogen detection, and the sensitivity of nematode identification is improved when the PCR amplification products are verified by an agarose gel electrophoresis instrument. The combination of PCR technology with other molecular biotechnologies expands the potential application of the PCR technology in the identification of specific species [2,34].

#### 4.2.1. Nested PCR (n-PCR)

In 2011, Huang devised a *B. xylophilus*-specific n-PCR protocol using topoisomerase I as the target and detected 44 nematode strains with obvious morphological characteristics, including *B. xylophilus*, *B. mucronatus*, *B. hofmanni*, *Seinura wuae*, *S. lii* and *Aphelenchoides macronucleatus* [35]. The n-PCR system can detect 50 fg of template DNA or an egg-sized nematode individual. It has high specificity and high sensitivity and analyses nematode samples extracted from wood infected by nematodes in the field. The advantage of n-PCR is its high specificity; even if the amount of DNA extracted from nematodes is very small, the specific region of *B. xylophilus* can be amplified by n-PCR. However, if an error occurs in the first amplification, the second amplification cannot be successfully implemented. Additionally, it is difficult to detect and distinguish multiple pathogens simultaneously using n-PCR [36]. Therefore, the use of n-PCR is time-consuming when a large number of nematodes need to be identified.

#### 4.2.2. Real-Time PCR (RT-PCR)

At present, RT-PCR is commonly used for the identification of nematode species [37]. By using the internal transcribed region of rDNA as a target and designing a pair of primers and a specific probe, the DNA content of *B. xylophilus* can be detected from at least 0.005 pg, and a single *B. xylophilus* individual can also be successfully detected [38,39,40]. RT-PCR has the advantages of being sensitive, reliable, safe and allowing a high throughput. The main disadvantages are that the experimental process is time-consuming, and the related reagents and equipment are relatively expensive. In 2009, the nematode innovation research team of Nanjing Forestry University further optimized the high-efficiency nematode lysate based on the original RT-PCR detection technology to directly cleave nematode DNA from pine samples. An automatic detection system for *B. xylophilus* was developed based on the automatic design of the detection process for specific nematode fragments, result interpretation and output reporting. Direct detection and automatic result interpretation of the presence of *B. xylophilus* in the sample can even be realized without separating the nematode from the diseased wood. The detection accuracy of the method approaches 100% when each gram of pine wood contains one *B. xylophilus*.

#### 4.2.3. Random Amplified Polymorphic DNA (RAPD)

The RAPD technology is a molecular technology based on PCR that can analyze the whole genome sequence. The interspecific and intraspecific differences and genetic diversity of *B. xylophilus* and *B. mucronatus* can be assessed with the RAPD-PCR technique to distinguish them [41,42]. According to the experimental results of Braasch et al., the primer OPY-01 distinguishes *B. xylophilus* and *B. mucronatus*, the primer OPB-7N distinguishes *B. xylophilus* and *B. fraudulentus*, and the primer OPPZ-08 is suitable for distinguishing *B. mucronatus* and *B. fraudulentus* [41]. Using the RAPD technology, Chen et al. screened the primer OPK09 and the primer combination OPC18+OPN18 from 140 random primers, and the two groups of primers were used to successfully distinguish *B. xylophilus* and *B. mucronatus* [43]. However, while the RAPD technology generates a large amount of information, it has a complex atlas, strict requirements for experimental reaction systems and conditions, and extremely sensitive amplification; moreover, its results often lack repeatability and comparability among laboratories, which leads to certain limitations when applying the RAPD technology for the identification of *B. xylophilus*.

#### 4.2.4. Sequence-Characterized Amplified Region (SCAR)

In the identification of nematodes, Meng et al. successfully transformed random amplified polymorphic DNA fragments specific to *Meloidogyne incognita* and *M. javanica* into SCAR markers with SCAR-PCR primers, and the amplification sensitivity in adults reached one-third of that in second-instar larvae [44]. Chen et al. successfully constructed a *B. xylophilus* detection kit using SCAR markers, with sensitivity reaching one-seventh of that for *B. xylophilus*, and achieved the accurate identification of the larvae [45]. The SCAR markers can quickly detect a large number of individuals, and the results exhibit good stability and high repeatability. However, this method has many programs, and it requires 2.5–3 h to complete the entire detection process.

### 4.3. Loop-Mediated Isothermal Amplification (LAMP)

At present, the LAMP technology is often used to detect and confirm the presence of *B. xylophilus* through the ITS region, the β-actin region, the *pel*-3 gene, the *syg*-2 gene and the expansin-like gene [46,47,48,49]. LAMP has the advantages of simple operation, low cost and low equipment demand and can be used for the on-site detection of fluorescent samples without expensive instruments and equipment. However, the products of LAMP amplification are not continuous fragments, which limits the application of cloning or other procedures. During the high-temperature and long-term amplification step, false positives may be obtained due to aerosol cross-contamination involving positive samples. In addition, a certain amount of error is introduced during the interpretation of color results.

### 4.4. Recombinase Polymerase Amplification (RPA)

Cha et al. designed primers for the 5S rDNA region of *B. xylophilus* and established the RPA amplification system. The system is capable of completing the exponential amplification of nucleic acids within 10 min, and the detection sensitivity is sufficient with 1.6 pg of *B. xylophilus* genomic DNA [50,51]. Thus, its detection efficiency is greatly improved compared with those of conventional PCR and LAMP techniques, but the synchronous detection of *B. xylophilus* and *B. mucronatus* has not been achieved. Based on previous studies, Fang et al. designed duplex-RPA primers using the ITS regions of *B. xylophilus* and *B. mucronatus* as targets, amplified the target nucleic acids within 30 min under a constant temperature of 37 °C and achieved the synchronous and rapid detection of *B. xylophilus* and *B. mucronatus* [52]. The POCD-RPA test method for the detection of *B. xylophilus* can facilitate the epidemiological investigation in this field as well as the quarantine process in the lumber industry. The RPA technology is relatively easy to use for people without nematode expertise, and the test results are accurate; however, special detection equipment is needed.

## 5. Detection of Pinewood Nematode Proteins

Protein-based methods have been widely used in nematode identification in recent years. Proteins play critical roles in living organisms, such as in cell proliferation and differentiation, energy conversion, or signal transduction [27]. Due to the redundancy of the genetic code, protein provides a smaller vocabulary than DNA, but the alphabet is much more complicated, with more than 20 characters, compared with four DNA bases. The distribution of membrane proteins is not uniform, and large differences exist between species, so proteins from different species act as a unique “bar code” that facilitates the identification of nematode species; interspecies differences can also be found by analyzing protein expression levels and expression patterns [53,54]. In addition, protein structure and posttranslational modifications increase their potential diversity and can therefore be used to define nematode species and facilitate their identification.

### 5.1. Two-Dimensional Gel Electrophoresis (2-DGE)

Two-dimensional gel electrophoresis, abbreviated 2-DE or 2-D electrophoresis, is a form of gel electrophoresis commonly used for the analysis of proteins [55]. Based on differences in the size and mass/charge ratio of protein molecules among nematode species, the species can be distinguished by the results of 2-DGE. After extracting proteins from *B. xylophilus* and *B. mucronatus*, Fu et al. compared the protein expression patterns between the two species by 2-DGE. Compared with *B. mucronatus*, *B. xylophilus* showed 15 highly expressed specific proteins [56]. The 2-DGE technique has the advantages of high speed and high resolution and it is also the only technology that can separate and display thousands of proteins at the same time. The disadvantages are also obvious: the number of proteins isolated and the results of polymorphism observed depend on the procedures used and the number of samples analyzed; it is sometimes difficult to distinguish protein spots because it is difficult to determine whether the observed similarities or differences are real or caused by gel deformation; and the procedure is more complex than others.

### 5.2. Isozyme Analysis

Isozymes are a type of proteins that are widely present in organisms. The main separation methods for isozymes are electrophoresis, chromatography, enzyme assays and immunology techniques, among which electrophoresis is the most widely used. After the soluble proteins are extracted from nematodes, the specific isozymes are stained. According to the molecular structure, activity and immunogenicity of the isozymes, different nematode species can be identified by polyacrylamide gel electrophoresis. Hu et al. performed enzyme electrophoresis of *B. xylophilus* and *B. mucronatus* [57]. The patterns of malate dehydrogenase, cellulose and esterase differed, but these differences were not stable; only glutamate oxaloacetate transaminase could successfully distinguish *B. xylophilus* from *B. mucronatus*. Although malate dehydrogenase and superoxide dismutase are currently used in the detection of nematode proteins, this method is tedious and time-consuming, and a positive sample must be available as a reference; therefore, its use is limited.

### 5.3. Matrix-Assisted Laser Desorption/Ionization–Time of Flight Mass Spectrometry (MALDI–TOF MS)

With this approach, specific peaks in the protein spectrum of different species or populations are used as biomarkers for protein identification. Zheng et al. used the MALDI–TOF MS technique to analyze the differentially expressed proteins in inoculated and unvaccinated sources of *B. xylophilus* within two weeks and successfully identified 87 differentially expressed proteins [58]. Fu et al. compared the protein expression of *B. xylophilus* and *B. mucronatus* by 2-DGE, analyzed 15 protein spots with different intensities by MALDI–TOF MS, and identified the specific proteins of *B. xylophilus* [56]. Luo et al. studied the protein differences between *B. xylophilus* and *B. mucronatus* by MALDI–TOF MS and identified 45 differential proteins, which laid a foundation for the accurate identification of *B. xylophilus* and *B. mucronatus* [59]. The combination of 2-DGE and MALDI–TOF MS provides a powerful tool for the classification of nematodes. This technique can not only identify nematode species but also detect whether pine trees are infected with *B. xylophilus*. However, the results are affected by many factors, including the method of protein extraction, the quality of the 2-DGE results and the accuracy of the instrument. Nematodes of different ages and grown under different conditions also exhibit different expression profiles.

## 6. Detection of Pine Volatiles

Plants emit many volatile organic compounds (VOCs) into the surrounding environment, which play an important role in their growth, communication, defense and survival [60,61]. VOCs released from the leaf surface are low-molecular-weight compounds with high vapor pressure and a low boiling point; these are the terminal metabolites of the host plant and can reflect its physiology and health status. They easily remain in the gaseous phase at standard temperature and pressure, usually at ultralow concentrations below the human olfactory threshold [62]. VOC analysis is a new field with great application potential that can be used to quickly, repeatedly and noninvasively monitor the health status of plants; based on this approach, diseases can be detected at different stages based on quantitative information collected from VOC samples [63,64]. VOC maps represent a new method of disease detection that can detect “plant-to-plant”, “plant-to-insect” and “plant-to-disease” communication mechanisms and obtain host markers for pathogens and abiotic stressors [65,66].

Gas chromatography–mass spectrometry (GC–MS) is one of the fastest growing and most dynamic techniques in metabonomic-based research [67]. When the PWN is artificially inoculated into pine trees, the inoculated pine trees stop exuding resin and subsequently release large amounts of ethylene or terpenoids [68,69]. Takeuchi et al. used GC–MS to examine *P. thunbergii* and found that the emissions of terpenoids, such as (-)-α-pinene, increased upon infection with PWD, and the timing of VOC increase overlapped with that of *M. alternatus* spawning [70]. Limonene and abietadiene are the main volatiles emitted by PWN-infected *P. pinaster* branches [71]. After inoculation with *B. xylophilus*, the 3-carene volatilization for *P. densiflora* and *P. koraiensis* was 9.7 and 54.7 times higher than that of healthy pine trees, respectively, indicating that this VOC can be used a marker of early infection with PWD [72]. However, the GC–MS analysis of volatile compounds is limited, in that impurities are easily introduced in the detection process, and the method has poor reproducibility. In practical applications, much time is spent cleaning and leaching the adsorption column and performing a desorption of the extraction head and sampling.

## 7. Detection via Spectral Techniques

In recent years, with the rapid development of geospatial information science and sensor technology, the potential for real-time and dynamic Earth observation at the macro scale has been significantly enhanced. The sky–Earth integrated Earth observation network formed by comprehensive ground surveys, satellite remote sensing and unmanned aerial vehicles (UAVs) has been widely used in the monitoring of geographical conditions and it has also been widely studied and validated in the field of PWD monitoring and detection [73].

### 7.1. Satellite Remote Sensing

Optical satellite remote sensing is multiband and multiphase, which are advantageous in monitoring, locating and evaluating discolored pine trees, and is one of the important means of forest pest-disease monitoring [74]. Lee et al. evaluated many approaches for detecting infected pine trees by using various remote sensing data (including high-spatial-resolution satellite images from 2000/2003 IKONOS and 2005 QuickBird, aerial photos and digital airborne data) [75]. Zhou et al. proposed an automatic method for identifying diseased trees based on convolutional neural networks and bounding box tools by classifying and identifying the remote sensing images acquired by high-resolution Earth observation satellites. This method can quickly locate epidemic areas, and the recognition accuracy of the test data sets approached 99.4% [76]. Zhang et al. proposed a detection method relying on spatiotemporal changes to identify trees infected by PWD using high-resolution remote sensing [77]. However, satellite remote sensing has certain limitations for detecting PWD, and it is difficult to capture detailed changes, especially when the number of infected trees in a forest is small [78].

### 7.2. Unmanned Aerial Vehicles (UAVs)

The diseased or dead pine trees infected by PWD are spatially dispersed, and it is difficult to use satellite remote sensing to monitor disease spread. Kim et al. used aerial photograph data with multitemporal hyperspectral 1 m spatial resolution from UAVs, the NDVI and VIgreen, to identify fallen and standing discolored pine trees [79]. Ding et al. based on the faster region convolutional neural network (Faster-RCNN) deep learning framework of the region proposal network (RPN) and ResNet residual neural network, used UAV remote sensing and artificial intelligence technology to train a PWD model. After network optimization, the detection accuracy was improved up to 90% [80]. Based on UAV remote sensing, Yu et al. adopted two target detection algorithms (Faster-RCNN and YOLOv4) and two traditional machine learning algorithms (random forest and support vector machine) for the early detection of infected pine trees [81]. However, currently, the interpretation of UAV remote sensing images still mostly relies on manual visual inspection, which requires skilled interpreters, has a high degree of subjectivity and poses challenges in establishing unifying standards through consensus and in meeting the requirements of a rapid interpretation of data from large areas. To some extent, this has delayed the clean-up of infected trees by forest rangers and accelerated the spread of the PWD epidemic.

## 8. Conclusions

In this paper, we reviewed the currently available, commonly used techniques for detecting PWD. The methods include the diagnosis of dead pine trees in forests, the morphological analysis of nematodes, the molecular detection of nematodes and infected trees and their detection based on pine volatiles and spectral techniques. The morphological identification of *B. xylophilus* has been the benchmark and basis for other detection methods. However, given the disadvantages of the morphological identification, PWD diagnosis has shifted from the original morphological method to others based on nematode molecular biology and chemical reactions of trees. The rapid and accurate diagnosis of PWD may prevent its further spread, but how to correctly and effectively integrate these detection methods is a topic that requires further study (Figure 1, Table 1).

The number of similar species and the availability of large samples play a role in the research and development of the molecular detection technology. The accuracy of detection also depends on the continuity of inspection in the subsequent process of practice testing. At the end of the 20th century, RFLP, RAPD, SCAR and other technologies advanced the species identification of PWNs, but these methods were not widely used in practice because of their complexity, lack of repeatability, operational procedures and lengthy detection time. In recent years, LAMP, RPA and other detection methods have advanced in terms of detection convenience, time and cost, but these methods still need to be tested in practice. The protein-based method represents a promising direction for future research on species identification. However, protein expression patterns are complicated, and the extracted proteins easily degrade, which may affect the accuracy of the assessment; this limitation is the main difficulty in applying this method.

In the initial stage of PWD infection, pine trees do not exhibit symptoms, so it is difficult to accurately determine whether the trees in a forest are infected. However, treatment is essentially impossible once symptoms related to the late stage of infection appear. Therefore, a new method for detecting VOC markers or analyzing the spectra of pine trees infected with PWD is needed. The epidemic areas of PWD are mainly distributed in the mountainous areas of Southeast China. These areas have complex terrain and dense forests, so it is very difficult to detect tree VOCs, and the quality of optical remote sensing images is greatly impacted by clouds. After pine trees are infected with PWD, their needles turn yellow, and the difference in hyperspectral reflectance is relatively easy to detect using remote sensing images. Therefore, the remote sensing technology is currently used in large-scale censuses and monitoring. With future scientific advances, a combination of detection methods may facilitate the detection of PWD.

## Figures and Tables

**Figure 1 ijms-23-10797-f001:**
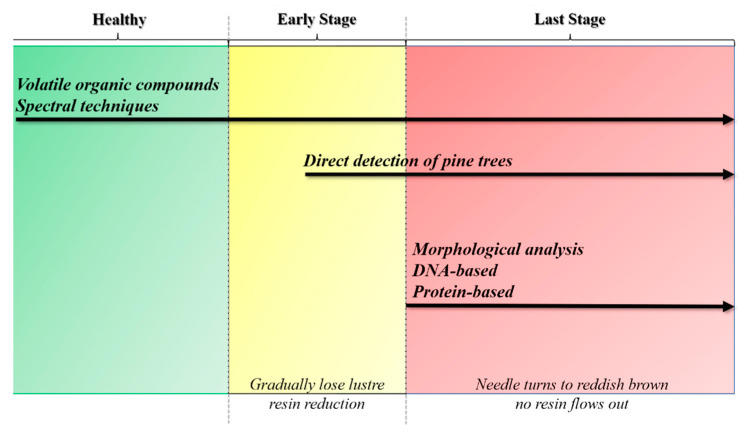
The timing of different methods for detecting pine wilt disease according to disease progression. Three disease stages are considered.

**Table 1 ijms-23-10797-t001:** Detection methods of pine wilt disease.

	Method	Sample	Advantages	Limitations	References
Direct inspection of pine trees	Manual check	Wild pine trees	Fast	Requires technical personnel expertise, possibility of subjective judgement	[17,18]
Morphological analysis	Microscope	Nematode	Low cost	Requires technical personnel expertise, possibility of subjective judgement	[19,22,24]
DNA-based methods	RFLP	Target DNA	Tool for analyzing the genetic variation of nematodes within and between species	Time-consuming, complicated, requires high-purity DNA	[28,29,30,31]
	n-PCR	Target DNA	High specificity	Time-consuming, difficult to detect and distinguish multiple species	[35,36]
	RT-PCR	Target DNA	Sensitive, reliable, safe and high throughput	Time consuming, equipment relatively expensive	[31,37]
	RAPD	Target DNA	Generates a large amount of information	Lacks repeatability, requires strict experimental reaction	[42,43]
	SCAR	Target DNA	High sensitivity	Time-consuming	[45]
	LAMP	Target DNA	Low cost, simple operation, low equipment demand	False-positive results	[46,47,48]
	RPA	Target DNA	Easy to use, results are accurate	Requires special detection equipment	[51,52]
Protein-based methods	2-DGE	Nematode protein	Fast, high resolution, can separate and display thousands of proteins at the same time	Time-consuming, difficult to distinguish the same protein spots	[56]
	Isozyme analysis	Nematode protein	High sensitivity	Tedious and time-consuming	[57]
	MALDI–TOF MS	Nematode protein	High sensitivity	Time consuming, requires specialized skills	[58,59]
VOC-based method	GC–MS	VOCs	Fast, allows repetitive, noninvasive, dynamic monitoring	Poor reproducibility	[70,71,72]
Spectral techniques	Satellite remote sensing	Wild pine trees	Fast, large-area detection, dynamic monitoring	Requires technical personnel expertise, difficult to capture detailed changes	[74,75,76,77]
	UAVs	Wild pine trees	Fast, large-area detection, dynamic monitoring	Requires technical personnel expertise	[80,81]

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
