# Peer review of "The Detection of Pine Wilt Disease: A Literature Review"

_ijms, 2022, doi:10.3390/ijms231810797_

Round 1
Reviewer 1 Report
Review paper: The detection of pine wilt disease: A literature review
General comments
This paper reviews the methods to detect the pine wilt disease PWD. PWD is a serious threat to pine trees and forests in the world and early detection and effective control measures are essential to better control its expansion. However, detecting individual trees infected as PWD is not practically straightforward. This paper give a comprehensive review on the way to detect PWD.
Based on literature, the authors introduce a wide range of methods to detect PWD; direct detection in the field, detection based on pinewood nematode morphology, detection of the nematode DNA, detection of the nematode protein, detection of pine volatiles that infected trees emit, detection by leaf spectral in terms of morphological, molecular-, chemical- and physical-based method. These methods cannot be covered by a single researcher and comprehensive approaches are necessary to better detect PWD.
This review paper is a nice contribution to those who work on PWD, I think, and it is worth to be published. Reading the paper, however, I feel that Conclusion is too plain and that the authors just list up various methods previous studies have used. I strongly suggest the authors to stress the importance to call for comprehensive approaches, in terms of macro and micro biology, and use of remote sensing and image analysis to develop a better way to detect PWD.
Specific comments
Because I am not molecular biologist, and not structural biologist, I could not properly evaluate the section 4. Detection of pinewood nematode DNA and 5. Detection of pinewood nematode protein. In these two sections, the authors list up various techniques focusing on DNA and protein with details. For evaluation of the descriptions in these sections, please refer to other reviewers who are specialized on molecule and protein studies.
Figure 3 is nice as it shows which techniques are most appropriate to detect PWD.
Conclusions (line 421-) is too plain. Please mention the authors' view about how various techniques can be used to detect PWD.
End of comments
Author Response
Responses to Reviewer 1:
Comments to the Author
1) This review paper is a nice contribution to those who work on PWD, I think, and it is worth to be published.
Response: Thank you very much for your patient review and comments.
2) Because I am not molecular biologist, and not structural biologist, I could not properly evaluate the section 4. Detection of pinewood nematode DNA and 5. Detection of pinewood nematode protein. In these two sections, the authors list up various techniques focusing on DNA and protein with details. For evaluation of the descriptions in these sections, please refer to other reviewers who are specialized on molecule and protein studies.
Response: Thank you very much for your patient review and comments.
3) Figure 3 is nice as it shows which techniques are most appropriate to detect PWD.
Response: Thank you for your affirmation.
4) Conclusions (line 421-) is too plain. Please mention the authors' view about how various techniques can be used to detect PWD.
Response: Thank you very much for your patient review and comments. We added the ‘Table 1’ to explain that.

Reviewer 2 Report
This manuscript is a review focused on several kinds of detection methods of pine wilt disease and the pathogen, Bursaphelenchus xylophilus, and contains useful information which merit publication in International Journal of Molecular Sciences. I have, however, some suggestions. These are given below.
1. Please check whether the references (e.g., No. 3) are appropriate and additional references (e.g., in L54-56) are needed in Introduction.
2. Figure 1 dose not correspond to the content in L29-32.
3. L44-46: Not Y. Tokushige but Y. Mamiya and T. Kiyohara (nematologists) identified the nematode as belonging to the genus Bursaphelenchus.
Mamiya, Y.; Kiyohara T. Description of Bursaphelenchus lignicolus n. sp. (Nematoda: Aphelenchoididae) from pine wood and histopathology of nematode-infested trees. Nematologica 1972, 18, 120-124.
4. L75-76: The meaning of this sentence is unclear.
5. Please explain the relationship between four characteristics in L81-85 and early and last stages in Figure 3.
Minor changes
L11: a summary the common techniques → a summary of the common techniques
L41: a national project → a national project in Japan
L126: larvae → juveniles
Reviewer 3 Report
Manuscript Number: ijms-1853100
Title: The detection of pine wilt disease: A literature review
Authors: Min Li, et al.
Remarks:
This submitted review paper explains and summarizes several procedures and techniques on detecting pine wilt disease (PWD), and discussed about their rapid and effective uses to reduce the damage caused by PWD. I think that the review makes the readers comprehend the history of studies on PWD detection and protection, and consider for the ongoing and future approaches and technologies for managements against PWD.
However, the present version of the paper needs some major revisions, because of unclear explanation of backgrounds (section 1), lack of visible and effective presentations using Tables, and also verbose and unnecessary sentences (in particular, sections 4-7) as commented below. These weak points make it difficult to tell the readers what the authors really want to say. I recommend the authors to reconsider carefully the presentation designs and resubmit, representing with more relevant logic, concise and catchy explanations.
Major Comments:
1. One serious unclearness of the present paper is the lack of tables which summarize the previous related studies of each technology topic systematically. Most of ‘review articles’ generally contain such tables representing insights or results of the previous studies with citations, which the readers can catch the general or specific knowledge and tendencies regarding with the particular topics. For example, the authors should summarize the characteristics or differences of several DNA markers (section 4) and protein-based procedures (section 5) containing each of their merits and demerits (time, accuracy, sample number, cost, and so on). I recommend the authors strongly to reconsider again the contents on the summary of the studies or technologies, according to what they want to tell the readers.
2. In a paragraph of Lines 68-78 in Introduction, the authors state that fast and accurate methods of the disease detection is important. In this paper, one of solution is to apply spectral and VOC techniques, which the authors say that diagnosis can be done before the visible symptoms appear (as in Fig. 3). If so, the statement like the end of this paper (two sentences of Lines 447-449: “In the initial stage, …”) should be moved into Introduction, as a background of needing development of the rapid detection technologies.
3. In sections 4 (nematode DNA markers) and 7 (spectral technique), the general explanations, regarding with the background and progress of each technology, and so on, are needed before individual concrete explanations (subsections), as done in the sections 5 and 6.
4. There are many unnecessary explanations and the paper has too volume. For example, in sections 4 and 5, explanation of each marker itself is unnecessary; such as Lines 132-140 of section 4.1: “Since the 1980s, … is revealed [27].”, Lines 149-160 of section 4.2: “Kary Mullis first proposed … of specific genes.”, Lines 171-179 of section 4.2.1: “In the amplification of … than the first.”, and so on. Every former sentences of each of the sub-(sub-)sections in the sections 4, 5, 6 and 7 seems unnecessary and do not relate directly to the main theme of the paper; the authors should concentrate on the markers’ application on the nematodes and the usages (merits or demerits), also using summary tables as wrote in the Major comment #1. Except for the points indicated as above, the authors should read again themselves and slim up the manuscript, particularly throughout the sections 4-7. Please reconsider.
Specific comments:
5. Lines 40-. What is “Later-”? Cut if unnecessary.
6. Lines 41-. Please explain what country the authors say about, such as “a national project (1968-1971)”, “pathologist Y. Tokushige observed …” (Lines 44-) and “in Kyushu” (Lines 47-)?
7. Lines 53-. Why not abbreviate “Bursaphelenchus xylophilus” only here?
8. Lines 115-. In this section 3, an explanation of extracting nematodes from trees is necessary, before “a traditional PWD detection” identifying nematode under a microscope. Is the ‘Baermann funnel’ a popular technique of the nematode detection?
9. Figure 3. I think that this has little information only on the applying stages of different methods (corresponding to the sections 2-7). It is better to convert into a table, together with additional information on the merits, usages or demerits for each of methods, and so on, as wrote in the Major comment #1. Please reconsider.
Author Response
Responses to Reviewer 3:
Comments to the Author
1) This submitted review paper explains and summarizes several procedures and techniques on detecting pine wilt disease (PWD), and discussed about their rapid and effective uses to reduce the damage caused by PWD. I think that the review makes the readers comprehend the history of studies on PWD detection and protection, and consider for the ongoing and future approaches and technologies for managements against PWD.
Response: Thank you very much for your patient review and comments.
2) One serious unclearness of the present paper is the lack of tables which summarize the previous related studies of each technology topic systematically. Most of ‘review articles’ generally contain such tables representing insights or results of the previous studies with citations, which the readers can catch the general or specific knowledge and tendencies regarding with the particular topics. For example, the authors should summarize the characteristics or differences of several DNA markers (section 4) and protein-based procedures (section 5) containing each of their merits and demerits (time, accuracy, sample number, cost, and so on). I recommend the authors strongly to reconsider again the contents on the summary of the studies or technologies, according to what they want to tell the readers.
Response: Thank you for your advice, the suggested table have been added.
Table 1. Detection methods of pine wilt disease.
|
Method |
Sample |
Advantages |
Limitations |
Reference |
Direct inspection of pine trees |
Manual check |
Wild pine trees |
Fast |
Requires technical personnel expertise, possibility of subjective judgement |
[17,18] |
Morphological analysis |
Microscope |
Nematode |
Low cost |
Requires technical personnel expertise, possibility of subjective judgement |
[19,22,24] |
DNA-based methods |
RFLP |
Target DNA |
Tool for analysing the genetic variation of nematodes within and between species |
Time consuming, complicated, requires high-purity DNA |
[29-32] |
|
n-PCR |
Target DNA |
High specificity |
Time consuming, difficult to detect and distinguish multiple species |
[37,38] |
|
RT-PCR |
Target DNA |
Sensitive, reliable, safe and high throughput |
Time consuming, equipment relatively expensive |
[32,39] |
|
RAPD |
Target DNA |
Generates a large amount of infor-mation |
Lacks repeatability, requires strict experimental reaction |
[44,45] |
|
SCAR |
Target DNA |
High sensitivity |
Time consuming |
[47] |
|
LAMP |
Target DNA |
Low cost, simple operation, low equipment demand |
False positive results |
[48-50] |
|
RPA |
Target DNA |
Easy to use, results accurate |
Requires special detection equipment |
[53,54] |
Protein-based methods |
2-DGE |
Nematode protein |
Fast, high resolution, can separate and display thousands of proteins at the same time |
Time consuming, difficult to distinguish the same protein spots |
[58] |
|
Isozyme analysis |
Nematode protein |
High sensitivity |
Tedious and time consuming |
[59] |
|
MALDI-TOF MS |
Nematode protein |
High sensitivity |
Time consuming, requires specialized skills |
[60,61] |
VOC-based method |
GC-MS |
VOCs |
Fast, allows repetitive, noninvasive, dynamic monitoring |
Poor reproducibility |
[72-74] |
Spectral techniques |
Satellite remote sensing |
Wild pine trees |
Fast, large-area detection, dynamic monitoring |
Requires technical personnel expertise, difficult to capture detailed changes |
[76-79] |
|
UAVs |
Wild pine trees |
Fast, large-area detection, dynamic monitoring |
Requires technical personnel expertise |
[82,83] |
3) In a paragraph of Lines 68-78 in Introduction, the authors state that fast and accurate methods of the disease detection is important. In this paper, one of solution is to apply spectral and VOC techniques, which the authors say that diagnosis can be done before the visible symptoms appear (as in Fig. 3). If so, the statement like the end of this paper (two sentences of Lines 447-449: “In the initial stage, …”) should be moved into Introduction, as a background of needing development of the rapid detection technologies.
Response: Thank you for your advice, this sentence has been moved to Introduction.
4) In sections 4 (nematode DNA markers) and 7 (spectral technique), the general explanations, regarding with the background and progress of each technology, and so on, are needed before individual concrete explanations (subsections), as done in the sections 5 and 6.
Response: Thank you for your advice, these backgrounds have been added.
e.g., 4. Detection of pinewood nematode DNA
Because of the increasing circulation speed of the wood market, the morphological identification method is far from meeting the demand. In recent years, with the continuous maturation and development of molecular biology technology, molecular detection methods for the PWN have been established and improved, and their application in practical work has become increasingly extensive[27]. This section focuses on a variety of molecular detection technologies developed based on DNA, and analyses their respective technical characteristics and application in the field of PWN detection, aiming at providing a reference for the development and application of new rapid molecular detection technologies for the PWN, which has very important theoretical significance.
5) There are many unnecessary explanations and the paper has too volume. For example, in sections 4 and 5, explanation of each marker itself is unnecessary; such as Lines 132-140 of section 4.1: “Since the 1980s, … is revealed [27].”, Lines 149-160 of section 4.2: “Kary Mullis first proposed … of specific genes.”, Lines 171-179 of section 4.2.1: “In the amplification of … than the first.”, and so on. Every former sentences of each of the sub-(sub-)sections in the sections 4, 5, 6 and 7 seems unnecessary and do not relate directly to the main theme of the paper; the authors should concentrate on the markers’ application on the nematodes and the usages (merits or demerits), also using summary tables as wrote in the Major comment #1. Except for the points indicated as above, the authors should read again themselves and slim up the manuscript, particularly throughout the sections 4-7. Please reconsider.
Response: Thank you for your advice, unnecessary explanations have been deleted.
6) Lines 40-. What is “Later-”? Cut if unnecessary.
Response: Thank you for your advice, this word has been deleted.
7) Lines 41-. Please explain what country the authors say about, such as “a national project (1968-1971)”, “pathologist Y. Tokushige observed …” (Lines 44-) and “in Kyushu” (Lines 47-)?
Response: Thank you for your advice, this sentence has been revised to:
However, a national project in Japan (1968-1971) implemented to control pine tree deaths, reported that these beetles never attacked healthy pine trees, only trees previously weakened by other biotic or abiotic factor; moreover, and the affected pine trees wilted before insect attack.
8) Lines 53-. Why not abbreviate “Bursaphelenchus xylophilus” only here?
Response: We’re very sorry about this, we have corrected this sentence:
- xylophilus can infect most Coniferales plants such as Pinus, Abies, Cedrus, Larix, Picea, Pseudotsuga and Tsuga species[4].
9) Lines 115-. In this section 3, an explanation of extracting nematodes from trees is necessary, before “a traditional PWD detection” identifying nematode under a microscope. Is the ‘Baermann funnel’ a popular technique of the nematode detection?
Response: We think that the Behrman funnel method is a general technique to isolate nematodes, not an identification technique, so we don't describe it.
10) Figure 3. I think that this has little information only on the applying stages of different methods (corresponding to the sections 2-7). It is better to convert into a table, together with additional information on the merits, usages or demerits for each of methods, and so on, as wrote in the Major comment #1. Please reconsider.
Response: Thank you for your advice, we have added the Tab.1, see in “Response 1”

Round 2
Reviewer 3 Report
Manuscript Number: ijms-1853100-v2
Title: The detection of pine wilt disease: A literature review
Authors: Min Li, et al.
Remarks:
The resubmitted review paper has improved substantially. It becomes to summarize effectively the development history of detection techniques of PWN. I consider that the paper is mostly acceptable for this journal. I comment on a few points which the authors should reconsider once again unnecessary sentences and conclusions.
Minor Comments:
1. Page 7. The starting sentences in subsection 4.1. (RFLPs) still have explanations of the marker itself which is not related with PWN diagnosis. “Due to base insertion … is revealed [28].” is unnecessary. Explanation on the use of RFLP to distinguish B. xylophilus (“Prior studies have …”) should be more important.
2. Page 8. Similarly, in subsection 4.2., the starting sentences of explanations of the PCR itself, “After dacades of development, … Since its inseption,” is unnecessary.
3. Page 17. I do not catch the last two paragraphs after Table 1. Are they in the Conclusions section? If so, they should be moved into Page 14.
Author Response
Comments to the Author
1) The resubmitted review paper has improved substantially. It becomes to summarize effectively the development history of detection techniques of PWN. I consider that the paper is mostly acceptable for this journal.
Response: Thank you very much for your patient review and comments.
2) Page 7. The starting sentences in subsection 4.1. (RFLPs) still have explanations of the marker itself which is not related with PWN diagnosis. “Due to base insertion … is revealed [28].” is unnecessary. Explanation on the use of RFLP to distinguish B. xylophilus (“Prior studies have …”) should be more important.
Response: Thank you for your advice, unnecessary explanations have been deleted.
3) Page 8. Similarly, in subsection 4.2., the starting sentences of explanations of the PCR itself, “After dacades of development, … Since its inseption,” is unnecessary.
Response: Thank you for your advice, unnecessary explanations have been deleted.
4) Page 17. I do not catch the last two paragraphs after Table 1. Are they in the Conclusions section? If so, they should be moved into Page 14.
Response: Thank you for your advice, we have moved the last two paragraphs into Page 14.